# On the Structural Peculiarities of Self-Reinforced Composite Materials Based on UHMWPE Fibers

**DOI:** 10.3390/polym13091408

**Published:** 2021-04-27

**Authors:** Dmitry Zherebtsov, Dilyus Chukov, Isabelle Royaud, Marc Ponçot, Ilya Larin, Eugene S. Statnik, Taisia Drozdova, Alexey Kirichenko, Alexey Salimon, Galal Sherif, Cyril Besnard, Alexander M. Korsunsky

**Affiliations:** 1Center for Composite Materials, National University of Science and Technology “MISiS”, Moscow 119049, Russia; Dil_Chukov@mail.ru (D.C.); elijah-larin@yandex.ru (I.L.); a.salimon@skoltech.ru (A.S.); eng_galal_emad@mu.edu.eg (G.S.); 2Institut Jean Lamour, Université de Lorraine, CNRS, IJL, F-54000 Nancy, France; Isabelle.Royaud@univ-lorraine.fr (I.R.); Marc.Poncot@univ-lorraine.fr (M.P.); 3Université de Lorraine, CNRS, LabEX “DAMAS”, F-57000 Metz, France; 4HSM Lab, Center for Energy Science and Technologies, Skolkovo Institute of Science and Technology, Moscow 121205, Russia; eugene.statnik@skoltech.ru (E.S.S.); alexander.korsunsky@eng.ox.ac.uk (A.M.K.); 5Technological Institute for Superhard and Novel Carbon Materials, Troitsk 108840, Russia; T.Shpitontseva@mail.ru; 6State Research Center of Russian Federation, Troitsk Institute for Innovation and Fusion Research, Troitsk 108840, Russia; akir73@mail.ru; 7Production and Design Department, Faculty of Engineering, Minia University, Minia 61111, Egypt; 8MBLEM, Department of Engineering Science, University of Oxford, Oxford OX1 3PJ, UK; cyril.besnard@eng.ox.ac.uk

**Keywords:** self-reinforced composites (SRCs), UHMWPE fibers, hot compaction, *Avizo*, Herman’s factor, Digital Image Correlation (DIC), *Ncorr*, *ImageJ*

## Abstract

The structure of self-reinforced composites (SRCs) based on ultra-high molecular weight polyethylene (UHMWPE) was studied by means of Wide-Angle X-ray Scattering (WAXS), X-ray tomography, Raman spectroscopy, Scanning Electron Microscopy (SEM) and in situ tensile testing in combination with advanced processing tools to determine the correlation between the processing conditions, on one hand, and the molecular structure and mechanical properties, on the other. SRCs were fabricated by hot compaction of UHMWPE fibers at different pressure and temperature combinations without addition of polymer matrix or softener. It was found by WAXS that higher compaction temperatures led to more extensive melting of fibers with the corresponding reduction of the Herman’s factor reflecting the degree of molecular orientation, while the increase of hot compaction pressure suppressed the melting of fibers within SRCs at a given temperature. X-ray tomography proved the absence of porosity while polarized light Raman spectroscopy measurements for both longitudinal and perpendicular fiber orientations showed qualitatively the anisotropy of SRC samples. SEM revealed that the matrix was formed by interlayers of molten polymer entrapped between fibers in SRCs. Moreover, in situ tensile tests demonstrated the increase of Young’s modulus and tensile strength with increasing temperature.

## 1. Introduction

The concept of self-reinforcement for composite materials was first to put forward by Capiati and Porter in 1975 [1]. Self-reinforced composite (SRC) materials based on polyolefins are well studied and reported in a range of reviews and books [2,3,4]. Nowadays, SRC materials based on polypropylene (PP) are commercially available, e.g., Curv^®^ [5], Armordone [6], PURE [7], etc. However, there are no commercially affordable SRC materials based on ultra-high molecular weight polyethylene (UHMWPE).

Polyethylene (PE) is considered to be more attractive for mechanical applications in most early studies since the theoretical elastic modulus of the linear PE molecule (~250 GPa) is much higher than the modulus of the helical PP molecule (~43 GPa) [4]. For this reason, it has been argued that the highest achievable modulus of the final structure is likely to depend on the characteristic properties of individual fibers, so that higher stiffness should be possible with PE fibers. The highest tensile strength and modulus of PE-based SRC were reported to be equal to 1.3 GPa and 73.9 GPa, respectively [8], while the same parameters for PP-based SRCs reached 0.17 GPa and 3.13 GPa, respectively [9]. Similar observations were made regarding the comparison of SRCs in terms of flexural properties [8,10].

The main, widely-expected advantage of SRCs is the guaranteed strong bonding between matrix and reinforcement components in this type of composite material that ensures good mechanical strength. The system can also be especially attractive for applications requiring chemical inertness. In addition, UHMWPE SRCs have lower density than traditional composites due to the lower density of polymeric reinforcing fibers compared to analogues such as carbon or glass fibers (approximately 1.0 g/cm^3^ versus 1.8–2.5 g/cm^3^). Moreover, SRCs should possess excellent recyclability due to their thermoplastic nature and the absence of the separation stage required to recycle traditional composites, although their recyclability has not been extensively studied previously.

Broadly speaking, the fabrication route of SRCs can be represented by two main concepts: (a) SRCs are manufactured from single starting polymer stock; and (b) SRCs are produced using distinct components of the same polymer type. The second concept allows achieving a wide temperature window that is important for stable properties of SRC’s end-product [11]. For this reason, there are numerous studies of SRCs based on the following composition: high-density PE(HDPE)/low-density PE (LDPE) [12], UHMWPE/HDPE [13], ultra-high-modulus PE/LDPE [14], etc. However, the difference in the polymer type can have negative effect on recyclability in further valuable applications. Besides, UHMWPE has the highest mechanical and tribological properties compared with LDPE and HDPE, so that the mixing of UHMWPE with other PE types decreases indicated properties. In addition, specific applications, e.g., in the biomedical field, require reinforcement using pure polymer: for instance, only pure UHMWPE is allowed for total joint replacements. Thus, the concept of pure polymer reinforcement is more attractive for medical and other applications.

Furthermore, it is important to compare mechanical properties of studied SRCs with other materials and their classes. For this reason, the logarithmic plot of specific stiffness versus specific strength is shown in Figure 1 to compare SRC based on UHMWPE fibers with other materials.

As evident from Figure 1, very few polymer-based composite materials can demonstrate the mechanical performance exceeding that of UHMWPE SRC’s studied in [16]. For example, unidirectional and biaxial composites made from epoxy resin and carbon fibers show superior specific strength compared to the SRC studied here—but not a combination of specific strength and specific stiffness. However, carbon fiber composites are difficult to recycle and cannot be used for medical applications. Another PE/PE composite found in Figure 1 was made by alternation of UHMWPE fibers and HDPE film. There are many applications that require polymer blending [17]. However, these composites have a limitation after recycling in areas where the pure polymer is required. Other materials are UHMWPE-oriented, highly-crystalline SPECTRATM fibers, which cannot be used as bulk material for structural applications. Worth to note that simple rule of mixture is valid for mechanical performance of UHMWPE materials when non-oriented matrix and highly-oriented fibers are taken as the constituents of a SRC composite promising further progress for SRCs of optimized structure and fiber content.

SRCs based on UHMWPE manufactured by hot compaction were studied since the end of 1990s until recently [11,18,19,20,21], and the influence of hot compaction conditions on the properties of SRCs has been reported. However, the processing—structure—properties—performance interrelation has not been understood fundamentally. In addition, we believe that this interrelation is crucial for performance optimization. In this article, we address the key issues of UHMWPE-based SRC manufacturing by reporting the data on morphology, crystallinity and Herman’s orientation factor determined by Wide-Angle X-ray Scattering (WAXS), X-ray tomography, polarized Raman spectroscopy, Scanning Electron Microscopy (SEM) methods and in situ tensile testing for quantitative evaluation of the impact of hot compaction conditions on the molecular orientation and properties.

## 2. Materials and Methods

### 2.1. Self-Reinforced Composite (SRC) Preparation

UHMWPE fibers from SGX (DSM Dyneema, Heerlen, Netherlands) with the average diameter of about 15–20 µm and a linear density of 220 dtex were used for the preparation of the SRCs. Hot compaction was used for producing rectangular samples with dimensions of 80 × 10 × 2 mm^3^. The initial fibers were wound unidirectionally between two fixed bobbins that were subsequently removed, and the wound fibers placed into the mold for hot compaction.

SRC samples were manufactured by hot compaction with a heating time of 50 min and molding time of 10 min followed by cooling in the mold to room temperature at molding pressure. The surface of each fiber undergoes partial melting during hot compaction followed by UHMWPE re-solidification during cooling. This re-solidified UHMWPE forms SRC matrix as illustrated in Figure 2. Samples were manufactured at temperatures of 145 °C, 155 °C or 165 °C to determine the dependence of structure on processing temperature. Different pressures of 25 MPa and 50 MPa were applied to determine the influence of this processing parameter on the SRC structure. In addition to standard conditions, hot compaction was carried out at two regimes: **Regime 1**—10 min at a certain molding pressure and temperature; **Regime 2**—8 min at molding pressure, then 2 min of release of molding pressure down to ambient pressure, and then return to molding pressure for immediate start of cooling. According to the Clausius–Clapeyron relation [21], elevated pressure shifts the melting temperature to a higher value. Conversely, pressure reduction promotes fiber melting.

### 2.2. Scanning Electron Microscopy (SEM) Characterization

Composite structural studies were conducted using JEOL JSM-6610LV SEM in the secondary electron (SE) imaging mode. The cross-sectional sample surfaces were prepared by cutting followed by polishing. After that, prepared samples were chemically etched to achieve contrast between molten and re-solidified UHMWPE and the unmodified fibers. The etching mixture contained one volume orthophosphoric acid per two volumes of sulfuric acid and 2% wt./vol. potassium permanganate [22]. Etching was carried out during 4 h under room conditions using orbital shaker. The etched surfaces were not treated by sputter coating, but SEM imaging was performed at the voltage of 15 keV under low vacuum conditions of 30–50 Pa chamber pressure and electron gun pressure not exceeding 5 × 10^−2^ Pa.

Histograms of the fiber cross-sectional size were calculated from obtained SEM images of samples manufactured using *ImageJ* software by employing the following approach: filtering → segmentation → statistics extraction. The median filter with a radius of 2.0 pixels was applied as a preprocessing step. Next, the morphological segmentation of the *ImageJ* plugin was used for fibers separation. Finally, the essential statistics like area and perimeter were extracted by analyzing the particles tool of the *ImageJ*. The example of applying the described technique to the SEM image is shown in Figure 3. The resulting histograms are shown and discussed in the next section.

### 2.3. Wide-Angle X-ray Scattering (WAXS) Analysis

WAXS in transmission mode was carried out to determine crystallinity ratio and crystalline chains orientation as a function of hot compaction conditions. Parabolic multilayer mirror (Osmic) X-ray optics was used together with a cylindrical capillary. The selected radiation wavelength was Cu K_α1_ (λ = 0.154 nm) generated using the voltage and current of 30 kV and 40 mA, respectively. The beam was collimated to the diameter of approximately 200 µm. 2D WAXS patterns were obtained with a sample to detector distance of 75 mm using a photosensitive image plate detector (FUJIFILM), and then read by a BASF 5000 scanner.

Considering the fiber cross-sectional size of ~15 µm, the signal collected corresponded to the average from the composite material, rather than its individual constituents. Herman’s factor *f* that is most often used to estimate the orientation of crystallites or molecules was calculated as a function of orientation for each determined lattice plane of each sample according to Equation (1) below. This value allows describing the orientation degree of objects of interest (for example, a normal of a crystallographic plane) relative to a chosen direction (for example, a fiber axis).
(1)f=12(3<cos2α(hkl,z)>−1)
where $<\cos^{2}\alpha_{\left(hkl,\;z\right)}>$ can be calculated as:(2)$$<\cos^{2}\alpha_{\left(hkl,z\right)}>=\frac{\int_{0}^{\frac{\pi }{2}}I\dot{\left(\alpha ,\theta \right)}\cdot \sin\alpha_{\left(hkl,z\right)}\cdot \cos^{2}\alpha_{\left(hkl,z\right)}d\alpha }{\int_{0}^{\frac{\pi }{2}}I\left(\alpha ,\theta \right)\cdot \sin\alpha_{\left(hkl,z\right)}d\alpha }$$
and I(α,θ) is the scattering intensity as a function of the azimuthal angle α and Bragg angle θ [23].

### 2.4. In Situ Tensile Test

The prepared samples were tested using a universal portable device as Deben Microtest 1 kN (The precision of force is 1 N). Tensile Stage (Deben UK Ltd., London, UK) in situ inside the chamber of the SEM Tescan Vega 3 (Tescan Orsay Holding, Brno, Czech Republic) in tensile mode. The illustration of the experiment is shown in Figure 4. Moreover, specimens with fiber winding in the transverse direction were sprayed with a thin Au layer about 5 μm to remove typical polymer charging during the SEM session. In contrast, samples with longitudinal to fiber direction were checked on air under room temperature. The indicated devices (SEM & Deben Microtest) were synchronized. The regime of the tests was as follows: (Backscattered-Electron) BSE imaging type, low vacuum mode at 10 Pa, resolution scan regime, HV 25 kV, magnification ×34, a field of view 6144 μm, speed scan 4 (3.20 μs/pixel), focus depth 2.940 mm for the SEM, and a permanent crosshead speed of 0.5 mm/min for Deben Microtest, respectively. To provide essential features on the sample’s surface for Digital Image Correlation (DIC) analysis, a glue containing Ag nanoparticles was used. The sequences of ~25 images were put into Matlab-based comprehensive and robust Ncorr program [24] and processed with the subset radius 29, subset space 2 and the number of threads 4. Video recordings of tensile tests were prepared for each specimen and uploaded as Appendix A with labels of the kind “Appendix A”, etc.

### 2.5. Raman Spectroscopy Measurements

Raman spectroscopy measurements were carried out to reveal the SRC anisotropy and to determine qualitatively the influence of hot compaction conditions on the SRC structure. Raman spectra were collected at room temperature using DXR2xi Raman Imaging Microscope (Thermo Fisher Scientific, MA, USA). The excitation source used was a monochromatic green line emitted by an Ar ion laser at a wavelength of 532 nm with a power of 10 mW and slit width of 50 µm. The SRC spectra were acquired by 200 scans for 0.1 s exposure time for each scan in the back-scattering geometry. Considering the beam spot diameter of 1 mm and fiber diameter of 25 µm, the averaged Raman scattering was obtained from the matrix phase and the reinforcing element, i.e., from the entire material.

Raman measurements for each sample were made at two positions of the initial beam polarization relative to the sample without using a polarizer for the response signal. In the first position, the polarization of the laser was parallel to the direction of the fibers in the sample (ψ = 0°) as shown in Figure 5. The other measurements were carried out with the polarization plane perpendicular to the axis of the sample (ψ = 90°).

### 2.6. X-ray Tomography Analysis

X-ray micro-computed tomography (μCT) measurements were carried out to prove absence of pores and to show the structure uniformity. Phoenix Nanotom X-ray μCT facility was used to characterize the morphology. Rectangular specimens with 8 × 2 × 2 mm^3^ in size were characterized with a resolution of 1 μm. The μCT procedure was based on the acquisition of a series of X-ray radiographs of a sample that was rotated step by step around a vertical axis perpendicular to the incident X-ray beam with the use of a copper filter. A mathematical algorithm was used to reconstruct the internal 3D volume structure of the samples. The final resolution of the 3D images was achieved to reach voxels of dimensions 2 × 2 × 2 μm^3^. Finally, to visualize and check the integrity of the structure for obtained datasets, *Avizo* software was used.

## 3. Results and Discussion

### 3.1. WAXS Analysis

SRC samples obtained at temperatures of 145 °C, 155 °C or 165 °C and pressures of 25 MPa or 50 MPa in both molding regimes during hot pressing were selected for WAXS characterization. Diffraction signal was acquired using photosensitive film, which accumulates X-ray signals diffracted from crystallographic planes. Exposure of about 15 h was applied to acquire 2D diffraction patterns of optimal quality (the intensity of the white color is proportional to the accumulated diffracted X-ray signal), which are shown, for example, in Figure 6.

Figure 6a demonstrates short and narrow segments of Debye rings corresponding to certain crystallographic planes of the orthorhombic O-phase. This manifests a number of features characteristic to the obtained UHMWPE SRCs: (a) high crystallinity; (b) predominant orientation of fibers along crystallographic directions and (c) high perfectness of crystalline phase. It also can be seen in Figure 6b,c that pattern segments tend to become longer and wider at higher compaction temperatures reflecting more pronounced melting that causes the misorientation of crystallites and perhaps the accumulation of defects.

1D diffraction patterns were extracted from 2D diffraction patterns for detailed analysis for 0° and 90° angles in respect to the main fiber axis as shown in Figure 7a. The extraction process involved symmetric sectors averaging (e.g., 0° and 180°) with a range from −5 to 5 degrees for two perpendicular directions. The intensity of X-ray peaks in 0–180° direction is clearly much weaker than in 90–270° direction. Further, the analysis was carried out for 1D X-ray diffraction spectra detected along 90° direction.

As demonstrated in Figure 7b almost no amorphous halo is revealed in the 1D X-ray diffraction spectra proving high degree of crystallinity in the studied SRC samples. An 1D spectrum contains a low intensity peak at approximately 2θ≈19.3° (corresponds to (010) reflection of monoclinic crystalline M-phase) and strong peaks at approximately 2θ≈21.5° and 24.1° (correspond to the reflections (110) and (200) of orthorhombic crystalline O-phase, respectively). Therefore, SRCs consists of mainly stable orthorhombic UHMWPE phase, and it also contains a small amount of metastable monoclinic phase that forms during the manufacture of fibers.

Processing of digital images [25] makes possible to analyze the variation of peak intensity with orientation angle over radial intensity distribution as exemplified in Figure 8a for each crystallographic plane of SRCs obtained at 145 °C and 25 MPa (Regime 1). The diffraction patterns appear in planes that are perpendicular to the indicated crystallographic planes. Thus, these signals form 90° with crystallographic planes. The strongest intensity of the main peaks was achieved in the vicinity of orientation angle of 90° and 270° for both O- and M-phases proving predominant orientation of crystallographic planes, crystallites and crystalline molecular chains along the fibers’ (and sample as prepared) principal axis.

The $<\cos^{2}\alpha_{\left(hkl,z\right)}>$ average cosine was calculated for the studied samples in accordance with equation (2) for each crystallographic plane, where α is the azimuthal angle between the sample axis and each detected plane considering 90° between the normals to planes and these planes. The values of Herman’s orientation factor *f* calculated using Equation (1) for each crystallographic plane for all SRC samples are given in Table 1.

As it seen from Table 1, Herman’s factors for both crystallographic planes in major O-phase are close to 1 revealing (a) a high degree of crystalline molecular chains orientation along the fiber axis and (b) almost no dependence of degree of orientation on temperature or pressure when hot compaction is carried out at Regime 1. Minor M-phase is less oriented and behaves similarly at compaction Regime 2. Under these conditions, the high degree of orientation of crystallites is preserved in UHMWPE SRCs even after hot compaction.

Herman’s factor turns out to be sensitive to temperature at lower compaction pressure 25 MPa demonstrating lower degree of orientation at higher compaction temperature. We interpret these data as the evidence of more pronounced melting and, as a result, better structural integrity due to the formation of matrix phase with lower degree of orientation than for the pristine fibers. The pressure release causes greater misorientation in SRCs in comparison with those obtained at constant molding pressure. Pressure release gives greater mobility to long polymer chains and through this more chances for intense amorphization of fiber surface at melting, i.e., the formation of less-oriented matrix.

In contrast, the elevated pressure allows preserving the orientation of crystallites and crystalline molecular chains due to the less-intense melting of pristine fibers. This interpretation was supported by the data from differential scanning calorimetry [20]. Overall, elevated temperature as well as pressure release result in the decrease of degree of orientation and, highly likely, in deterioration of mechanical performance.

### 3.2. Mechanical Perfomance

The specimens with fibers oriented longitudinally and transversely in respect of tension axis were tested to study the influence of hot compaction conditions on mechanical performance of SRC. The data on ultimate tensile strength (UTS) are presented in Table 2. The UTS_long_ in longitudinal direction scaled from 67.8 MPa to 174 MPa with the increase of compaction temperature from 145 °C to 155 °C that is believed to be caused by the increment of matrix volume, providing better load transfer to the reinforcing fibers. However, further increase of compaction temperature up to 165 °C led to decrease in strength down to 142.3 MPa due to the smaller ratio of fiber-to-matrix volumes. In contrast, UTS_trans_ in the transverse direction gradually grows from 4.1 MPa to 16.5 MPa with compaction temperature enhancing the volume fraction of isotropic molten PE matrix, since the mechanical load is transferred through the matrix when a unidirectional composite is extended in the transverse direction. Similar tendency for both directions was observed for unidirectional PE-based SRCs at bending test [21].

As expected, according to the Clausius–Clapeyron relation [26], elevated hot compaction pressure (50 MPa against 25 MPa) resulted in lower matrix content and, therefore, poorer UTS in both longitudinal and transverse directions for samples obtained at 155 °C, with UTS_long_ dropping from 174.3 MPa to 129.5 MPa (by −25,7%), and UTS_trans_ by −44,1%. On the other hand, pressure release seems to promote UTS_trans_—that is slightly increased from 7.7 MPa to 8.3 MPa. These facts are explained by more pronounced melting during hot compaction at pressure release that is also proven by WAXS analysis showing large misorientation for samples obtained with pressure release.

A series of transverse fiber winding specimens for SEM in situ tensile testing was fabricated under following conditions: molding temperature 150, 160 and 170 °C; molding pressure 25 (and 50 MPa for 160 °C). The results showed the same trends as described above. The loading curves for this series are represented in Figure 9.

The strain was taken from DIC analysis, where the program calculates it according to the Eulerian–Almansi theory. DIC analysis reveals significant strain localization in the middle of ROI. Video recordings (Appendix A) illustrate the character of structural damage and disintegration through fiber separation process (Figure 10).

### 3.3. SEM Characterization

Cross-sections of SRCs compacted at 150 °C, 160 °C and 170 °C (25 MPa) at Regime 1 are shown in Figure 11. The creation of relatively uniform monolithic structure is seen clearly, with fiber bonding via the formation of narrow interfiber matrix layer consisting of recrystallized molten polymer. Since the etching agent applied preferentially attacks amorphous isotropic UHMWPE, its location is revealed by the etching grooves. Therefore, it is concluded that only fiber surface experiences melting during hot compaction. Furthermore, the core regions of fibers are deformed to become polygonal at high pressure. It can also be assumed that the thickness of etched grooves correlates with the matrix volume fraction, gradually increasing with compaction temperature.

Statistical parameters for achieved cross-sections such as fiber size distribution for samples with different process conditions are shown in Figure 12. It should be mentioned that distributions were plotted for samples prepared under constant pressure. The analysis gave mean fiber sizes of 10, 20 and 24 μm for 150, 160 and 170 °C, respectively. The histogram obtained for the structure prepared at 150 °C has a lognormal distribution, while other plots do not reveal statistical significance for commonly anticipated distributions like normal of lognormal.

### 3.4. Raman Spectral Analysis

Raman spectroscopy measurements were carried out to assess qualitatively the anisotropy of SRC samples. The degree of orientation can be evaluated with change in vibrational bands intensity with rotation of the polarizer angle relative to the sample axis [27,28]. Raman spectra of SRC samples obtained at 165 °C and 25 MPa or 50 MPa (Regime 1) at 0° and 90° between the SRC axis and initial beam polarization are depicted in the Figure 13. For all samples, vibrational bands at 1130 cm^−1^ (C–C stretching bonds) and 1416 cm^−1^ (–CH_2_– bending vibrations in crystalline phase) became lower when the polarization angle changed from 0° to 90°. This is caused by the parallelism of the *c*-axis of UHMWPE orthorhombic crystallites, which are believed to be collinear to the fibers’ axis. A reverse trend is found for the bands at 1440 cm^−1^ and 1460 cm^−1^, which are likely corresponding to –CH_2_– bending vibrations in amorphous phase. The same result was observed for other types of specimens.

### 3.5. X-ray Tomography Analysis

X-ray tomography was carried out to visualize the possible internal defects (voids, cracks and other imperfections) and general 3D appearance of internal volumes of SRC samples, as represented in Figure 14. It was initially expected that the vanishing difference in density of matrix and reinforcing fibers would not permit to segment these constituents. Indeed, X-ray tomography demonstrates the structural integrity of the entire SRCs composite and, moreover, proves the lack of porosity or interfaces at achieved voxel size. Delamination can be detected in the sample corners where knife sawing, perhaps, detrimentally damages the material crushing more fragile matrix. This type of defects is not formed when gentle cutting with a diamond blade is applied.

## 4. Conclusions

Self-reinforced composites fabricated through hot molding of highly crystalline UHMWPE fibers promise the achievement of attractive combination of high mechanical performance, sustainability and chemical resistance applicable in many traditional engineering applications and biomedical devices. Optimization of technological parameters for a material obviously requires a thorough structural study with a variety of traditional and recently developed characterization techniques. This evokes revisiting and reviewing of materials science basics as undertaken in this research devoted to UHMWPE-based SRCs.

The achieved study showed an almost full spectrum of peculiarities of the self-reinforced composite structures based on UHMWPE manufactured for various preparation conditions that potentially have a wide range of future applications.

In this study, advanced and comprehensive methods like WAXS, X-ray tomography, SEM, mechanical test and Raman spectroscopy with a combination of modern processing tools were directed to precisely characterize the inner structure of the self-reinforced composites based on UHMWPE fibers produced for different processing conditions, namely, temperature and pressure, on micro and nanoscale levels. We found that manufactured SRCs have structural integrity without the presence of any other substances or chemical modifications that were observed by X-ray tomography and Raman spectroscopy methods. Besides, WAXS analysis demonstrated that a high degree of orientation of crystallites and crystalline molecular chains along the fiber axis is preserved in UHMWPE SRCs even after hot compaction and the current statement does not depend on the pressure and temperature. Moreover, SEM in situ mechanical tests showed that growth in hot compaction pressure resulted in lower matrix content and poorer longitudinal and transverse tensile strength that corresponds with the Clausius–Clapeyron relation.

To sum up, the optimal process condition of SRC preparation could be chosen as 165 °C under 25 MPa. The achieved results set the next investigation plan. For instance, SRCs could withstand carbon fibers and be applied for ballistic as an armor vest or used in medicine as a knee replacement implant due to their biocompatibility. So, it would be attractive to study the planting and winding of fibers’ impact on the composite properties and tribological response.

## Figures and Tables

**Figure 1 polymers-13-01408-f001:**
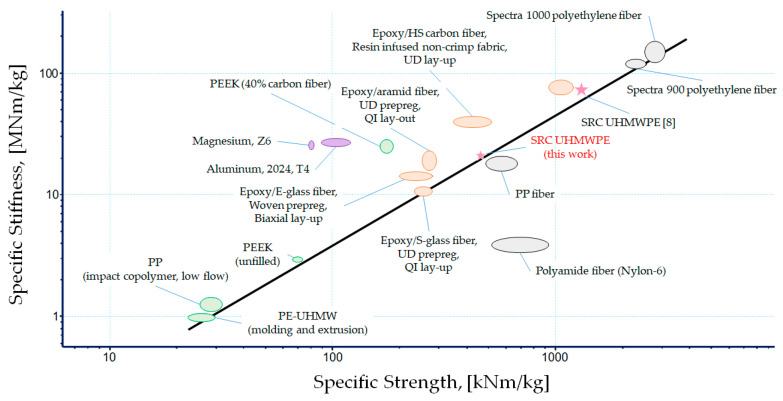
Ashby plot for various materials illustrating the range in specific strength and specific stiffness [15].

**Figure 2 polymers-13-01408-f002:**
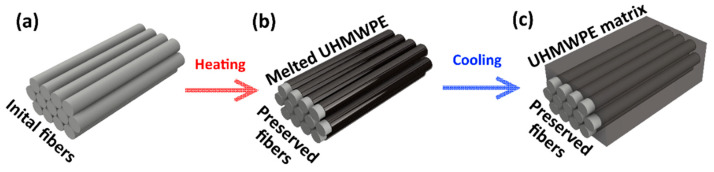
Schematic diagram of the self-reinforced composite (SRC) preparation: (**a**) initial fiber with winding, (**b**) molten fibers during hot compaction, and (**c**) the resulting self-reinforced composite.

**Figure 3 polymers-13-01408-f003:**
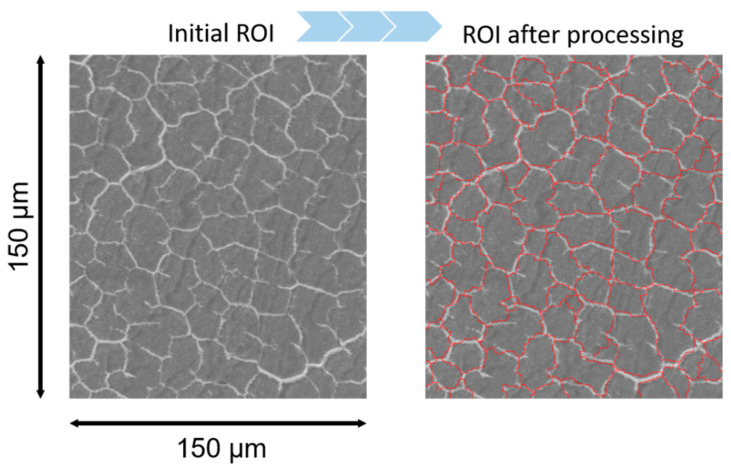
The result of applying the image processing technique for the SEM image of SRC by the *ImageJ* software.

**Figure 4 polymers-13-01408-f004:**
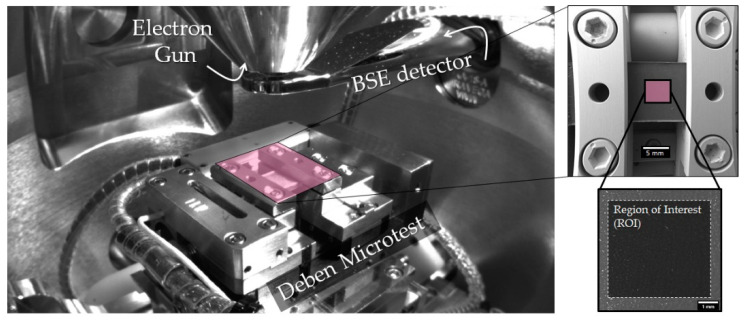
The typical illustration of in situ tensile test with the used Region of Interest (ROI) for DIC analysis.

**Figure 5 polymers-13-01408-f005:**
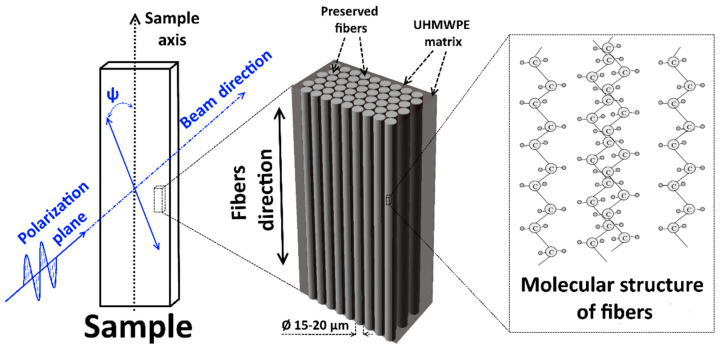
Position of the initial beam polarization relative to the sample, fibers and molecules.

**Figure 6 polymers-13-01408-f006:**
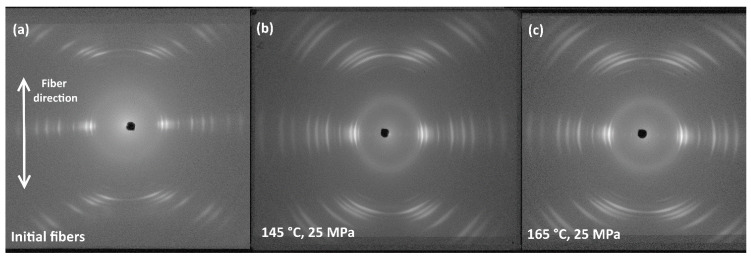
2D WAXS patterns of (**a**) the pristine fibers and samples made at 25 MPa and (**b**) 145 °C and (**c**) 165 °C (Regime 1).

**Figure 7 polymers-13-01408-f007:**
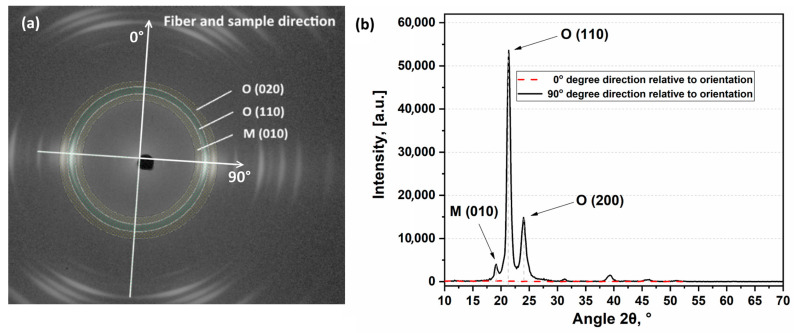
(**a**) 2D diffraction pattern and (**b**) 1D profile derived from 2D diffraction pattern—sample prepared at 145 °C and 25 MPa (Regime 2).

**Figure 8 polymers-13-01408-f008:**
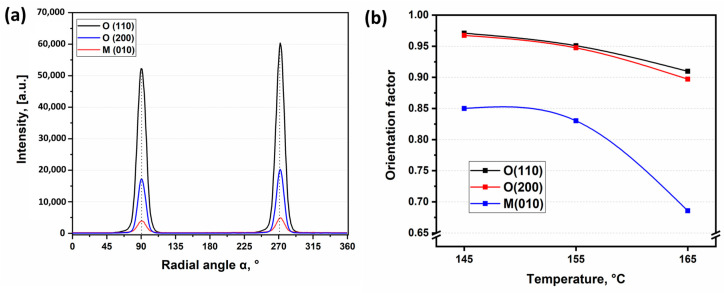
Radial intensity distribution for a composite obtained at 145 °C and 25 MPa (Regime 1) (**a**) and Herman’s factor (**b**) for a composite obtained at different temperatures and 25 MPa (Regime 1).

**Figure 9 polymers-13-01408-f009:**
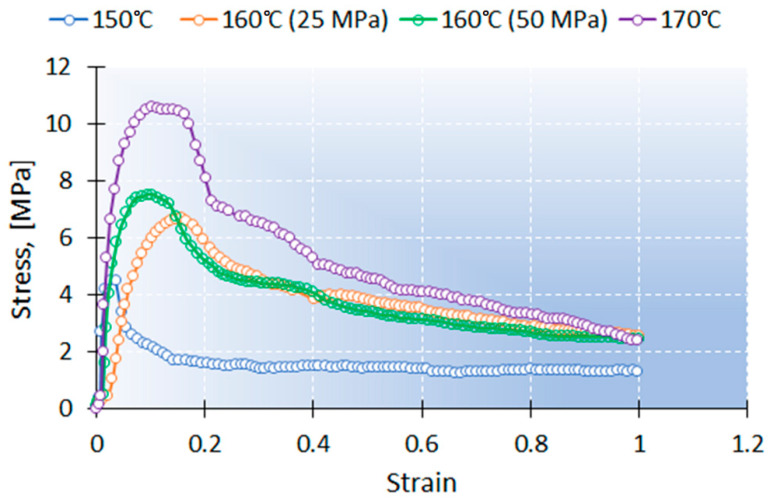
The loading curves for transverse fiber winding samples (molding temperature 150, 160 and 170 °C; molding pressure: 25 and 50 MPa).

**Figure 10 polymers-13-01408-f010:**
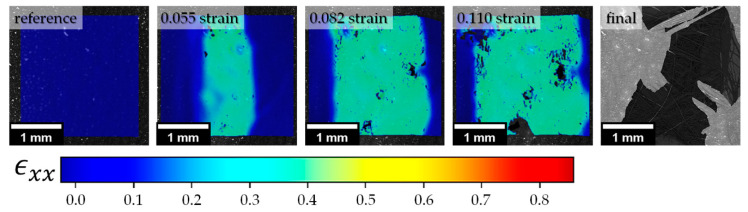
Typical strain localization and structure fluffiness observed by DIC and SEM, respectively, during in situ tensile test of SRCs with transverse fiber winding.

**Figure 11 polymers-13-01408-f011:**
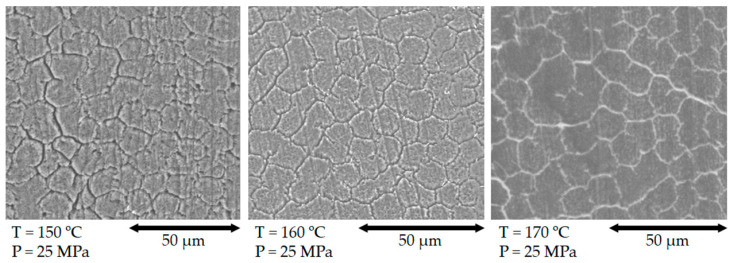
Cross-section of samples obtained 150 °C, 160 °C and 170 °C (25 MPa) (Regime 1).

**Figure 12 polymers-13-01408-f012:**
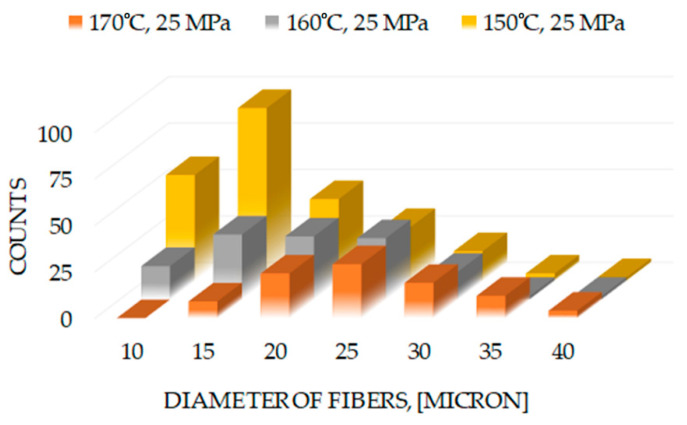
The histograms of fiber size distributions for samples with different preparation conditions.

**Figure 13 polymers-13-01408-f013:**
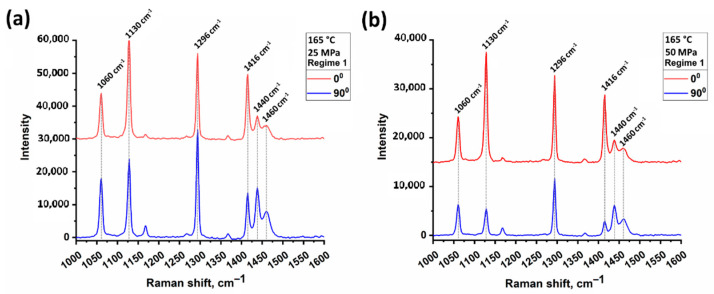
Raman spectra of SRCs obtained at 165 °C and 25 MPa (**a**) or 50 MPa (**b**) (Regime 1) at 0° (red) and 90°C (blue) between the SRC axis and initial beam polarization.

**Figure 14 polymers-13-01408-f014:**
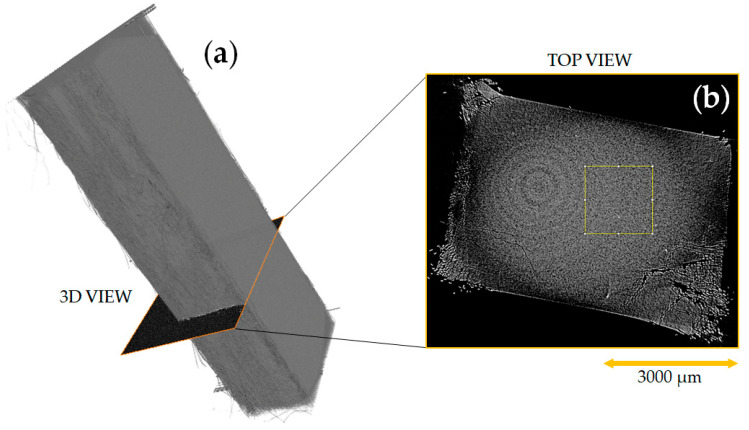
X-ray tomography (**a**) and cross-section (**b**) for SRC obtained at 165 and 25 MPa (Regime 1).

**Table 1 polymers-13-01408-t001:** Herman’s factor *f* for crystallographic planes for SRCs obtained at different conditions.

Regime 1						
T, °C	25 MPa			50 MPa		
	O (110)	O (200)	M (010)	O (110)	O (200)	M (010)
**145 °C**	0.971	0.968	0.869	0.97512	0.97242	0.877
**155 °C**	0.960	0.957	0.859	0.96178	0.95786	0.863
**165 °C**	0.960	0.954	0.856	0.96097	0.95671	0.859
**Regime 2**						
**T, °C**	**25 MPa**			**50 MPa**		
	**O (110)**	**O (200)**	**M (010)**	**O (110)**	**O (200)**	**M (010)**
**145 °C**	0.971	0.967	0.850	0.971	0.969	0.852
**155 °C**	0.950	0.947	0.830	0.958	0.952	0.837
**165 °C**	0.909	0.897	0.685	0.952	0.950	0.778

**Table 2 polymers-13-01408-t002:** Ultimate Tensile strength of SRCs in longitudinal and transverse directions.

Conditions			UTS_long_, MPa	UTS_trans_, MPa
T, °C	Pressure	Regime		
145 °C	25 MPa	1	67.8	4.1
155 °C	25 MPa	1	174.3	7.7
155 °C	25 MPa	2	164.7	8.3
165 °C	25 MPa	1	142.3	16.5
155 °C	50 MPa	1	129.5	4.3

## Data Availability

Data available on request due to restrictions of privacy.

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
