# Peer review of "On the Structural Peculiarities of Self-Reinforced Composite Materials Based on UHMWPE Fibers"

_polymers, 2021, doi:10.3390/polym13091408_

Round 1

Reviewer 1 Report

The paper is an interesting work focused on the characterization of self reinforced polymers. The work is carried out appropriately

Author Response

Dear reviewer, thank you for carefully reading the paper and giving good feedback!

Reviewer 2 Report

The article (Polymers-1187085) contributed by Zherebtsov et al entitled “On the Structural Peculiarities of Self-Reinforced Composite Materials based on UHMWPE Fibers” focuses on the structural characterization of self-reinforced composites of UHMWPE by means of a series of experiments. It was found that the inner structure of the composites varies with the processing conditions, which furtherly shows a different mechanical property. Overall, the work is decently written and the results are well presented. By judging the works presented by the authors, this reviewer thinks it is suitable for the journal of POLYMERS. However, for the benefit of the readers, the reviewer suggests addressing the following concerns prior to publication.

  1. In the section of abstract and conclusion, authors mentioned, “combination of modern processing tools as Avizo, ImageJ, Ncorr”. However, these software are very common, in this reviewer’s eyes, it is not necessary to emphasis these attached processed tools.
  2. In the abstract section, authors mentioned “cannot be broadly used after recycling due to the formation of a blend”(L95). This reviewer does not somewhat agree with this sentence. In fact, as a blend, it may limit the application in some cases. However, there are many applications existed which is caused by the blending. Herein, for extending the range of knowledge and benefiting with the reader, an article, Journal of Materials Engineering and Performance, 2017, 26(8), 4072-4082, which focuses on the topic of self-reinforcement using the PP/PE blends, was suggested to be mentioned.
  3. In the section of abstract, authors mentioned “the processing – structure – properties – performance interrelation that is crucial for performance optimization has not been …” (L104). This sentence doesn’t seem smooth.
  4. In L123, please carefully revise the sentence “… cooling to form …”.
  5. Many obvious mistakes can be found easily. For example, the equation (1), L176, L177, L 178, L257, L259, L280, etc.
  6. In L305, “in poorer mechanical performance” is somewhat not suitable. Maybe, the deterioration of mechanical performance?

In addition, a revision is suggested before its publication.

Author Response

Dear reviewer, thank you for carefully reading of the paper and your helpful suggestions. We corrected the paper according to your comments; see below for a more detailed description.

  • We deleted emphasized tools’ names.

  • We absolutely agree with your comment. We have added that a polymer blend cannot be used in particular cases. As you suggested, the article was mentioned in our article.

  • We corrected this sentence trying to preserve our main idea.

  • We also have tried to correct gently this sentence.

  • We are so regret that these mistakes were in our article. Unfortunately, we could not control it because these question-marks appeared in uploading stage of article. No any mistakes were noted in original files. We will carefully check it in next submission.

  • Thank you for your helpful suggestion. It seems better. We corrected it.

Round 2

Reviewer 2 Report

The revised version is better than before. I recommend to accept it in its present form.